# OpenReview forum: "Pruning Spurious Subgraphs for Graph Out-of-Distribution Generalization"
_NeurIPS.cc/2025/Conference — NeurIPS 2025 poster_

### Official Review · Reviewer_S6ZW · 2025-07-02

**Clarity:** 3
**Significance:** 3
**Originality:** 3
**Rating:** 4
**Confidence:** 3

**Summary:**

The paper proposes a graph OOD generalization algorithm via graph pruning. The motivation is that the key of OOD generalization is to identify the invariance parts (subgraphs) but some sprious edges prevent the model find the true invariant subgraphs. Thus, the authors want to prune these edges to let the model better understand the invariances in the graph.

**Questions:**

1. Do the authors can provide an in-depth analysis to show whether the sprious edges would be the key limitation in identifying the invariant subgraphs and why the edge pruning be the most suitable strategy?

**Ethical Concerns:**

["NO or VERY MINOR ethics concerns only"]

**Final Justification:**

This work provides a novel method in solving graph OOD problem. It shows an elegant method with strong theoretical basis, with desirable performance. I think the paper is above the bar of the NeurIPS.

**Limitations:**

See weaknesses.

**Quality:**

3

**Strengths And Weaknesses:**

Strengths:
1. The paper is easy to follow and well-written.
2. The motivation of pruning sprious edges seens to make sense.
3. The theoretical analysis further reveals the superiority of the method.

Weaknesses:
1. Despite the authors claim sprious edges is a significant issue in identifying key subgraphs, they do not provide an analysis on how and why it happen.
2. Following the previous one, the authors also do not clearly answer why the edge pruning could be suitable to handle the issue. Do there have any other strategies?

---

> ### Author Rebuttal · Authors · 2025-07-30
>
> Thank you for your constructive feedback! Please see below for our responses to your comments and concerns.
>
> ---
>
> > **W1: How and why spurious edges arise.**
>
> __Response:__ We appreciate the reviewer’s question. Spurious edges occur **naturally** in many real‑world domains due to the way data are collected or generated. Below we provide three concrete examples to illustrate this phenomenon:
>
> - **Molecular property prediction (e.g., GOOD-HIV [1]).** In molecular graphs, nodes represent atoms and edges represent chemical bonds. The *invariant* subgraphs are functional groups (e.g., –OH, –COOH) whose chemical bonds (edges) causally drive HIV inhibition. The *spurious edges* come from bonds forming large ring scaffolds or linking atoms that increase overall molecular size. These edges co‑occur with labels in training data because scaffolds and larger molecules are overrepresented among active compounds, even though these bonds are not causally responsible for inhibition.
>
> - **Social‑media analysis (e.g., Twitter15/16, PHEME [2]).** In rumor‑propagation graphs, nodes are users and edges represent retweet/reply interactions. The *invariant* structures are propagation paths (edges) that genuinely reflect veracity‑related user engagement. By contrast, *spurious edges* are event‑specific retweet links that form distinctive motifs, such as in PHEME’s Hurricane Sandy thread where retweet edges from `@WeatherNews` to `@StormTracker` and `@CityNews`, and then to `@LocalReporter`, form a Y‑shaped cascade. These edges are unique to that event and spuriously correlate with rumor labels without causal relation to truthfulness.
>
> - **Brain‑network studies (resting‑state fMRI [3,4]).** In functional connectivity graphs, nodes are brain regions and edges represent statistical correlations between BOLD (Blood‑Oxygen‑Level‑Dependent) signals. The *invariant* edges correspond to genuine neural interactions. The *spurious edges* are those inflated by head‑motion artifacts, such as correlations between frontal poles caused by micromovements. These edges often co‑vary with demographic or diagnostic variables, misleading models into associating motion‑induced noise with disease labels.
>
> These examples show that spurious edges are not artificially imposed but are an inherent **byproduct** of real‑world data collection processes, which further justifies our pruning‑based approach for graph OOD generalization.
>
> To be more self‑contained, we have added a discussion in the appendix explaining why and how spurious correlation arise and how they impact OOD performance.
>
> > **W2: Why edge pruning is suitable.**
>
> **Response:** The key reason why edge pruning is suitable for OOD generalization is that, as shown in Section 3 of our paper, state‑of‑the‑art methods that directly attempt to identify invariant edges are error‑prone: spurious edges with strong label correlations are often **misclassified** as invariant, leading to incomplete or incorrect estimation of the true invariant subgraph. In contrast, pruning a subset of spurious edges provides a more reliable way to preserve $G_c$, making it easier to retain the invariant structure. This is why pruning spurious edges is particularly suitable for enhancing OOD generalization ability.
>
> Indeed, there exist many other strategies and methods in the graph OOD literature that address this challenge through alternative strategies, including supervised contrastive learning, data augmentation, information bottleneck, and other regularization‑based techniques. However, all of these methods focus on directly identifying invariant edges; none of them explicitly leverage **edge pruning** as a means to improve OOD generalization. PrunE is the first to explore this pruning‑based paradigm, and both our theoretical analysis and empirical results demonstrate its effectiveness compared to prior approaches.
>
>
> > **Q1: The necessity of pruning spurious edges.**
>
> __Response:__ We thank the reviewer for this perceptive question. Please see our response below:
>
> ### **1. Spurious edges as the core limitation**
>
> It is widely acknowledged that *spurious subgraphs* are a primary driver of poor generalization in graph OOD literatures. For instance, in molecular property prediction, GNNs often exploit ring scaffolds or overall molecule size for prediction, which correlate with training labels but lack causal effect on target labels.
>
> As demonstrated in Section 3 of our draft, state‑of‑the‑art methods that attempt to *identify invariant edges* tend to misclassify spurious edges as “causal edges”, leading to incorrect subgraph estimation and degraded OOD performance. This implies that spurious edges are the key limitation for current SOTA methods to estimate invariant subgraphs accurately.
>
> ### **2. Pruning as the most suitable strategy**
>
> While it is challenging to rigorously prove that pruning is the most effective (optimal) way, our experiments show that PrunE significantly outperforms a broad range of general and graph OOD approaches that seek to directly identify invariant edges, on both synthetic and real-world splits. Moreover, our OOD regularization is conceptually simple and insensitive to hyperparameter variations, further underscoring the potential and practical appeal of the pruning paradigm.
>
> Together, these points illustrate why **pruning spurious edges** is especially effective in OOD settings for graph-structured data.
>
>
> ---
>
> We sincerely thank the reviewer for the careful review and insightful feedback. We hope that our responses have adequately addressed your concerns regarding our study. Please let us know if there is anything we could further discuss, your further feedback would be greatly appreciated.
>
>
> **References**
>
> 1. Gui, et al., GOOD: A Graph Out-of-Distribution Benchmark. NeurIPS 2022.
>
> 2. Wu, et al., Probing Spurious Correlations in Popular Event-Based Rumor Detection Benchmarks, arXiv:2209.08799.
>
> 3. Satterthwaite, et al., Motion artifact in studies of functional connectivity: Characteristics and mitigation strategies, Human Brain Mapping.
>
> 4. Xu, et al., BrainOOD: Out-of-distribution Generalizable Brain Network Analysis. ICLR 2025.

---

> > ### Comment · Reviewer_S6ZW · 2025-08-05
> >
> > The authors have resolved by concerns. I'd like to keep my positive rating.

---

> > > ### Author Response · Authors · 2025-08-06
> > >
> > > Thank you for your valuable feedback! We are glad to hear that the concerns have been addressed.

---

### Official Review · Reviewer_hmUG · 2025-07-03

**Clarity:** 2
**Significance:** 2
**Originality:** 3
**Rating:** 4
**Confidence:** 2

**Summary:**

The paper introduces PrunE, aiming to improve out-of-distribution (OOD) generalization in Graph Neural Networks (GNNs). Specifically, instead of directly identifying invariant subgraphs, PrunE, focuses on pruning spurious edges from the graph, employing two main regularization terms, i.e., graph size constraint and probability alignment: Forces the lowest K% of edge probabilities toward zero, further reducing the presence of spurious edges. Theoretical analysis and extensive experiments on both synthetic and real-world datasets seem to demonstrate the effectiveness of the proposed method.

**Questions:**

Please refer to my summary of weakness.

**Ethical Concerns:**

["NO or VERY MINOR ethics concerns only"]

**Final Justification:**

I have carefully read the rebuttal and will maintain my positive score.

**Limitations:**

yes

**Paper Formatting Concerns:**

No.

**Quality:**

3

**Strengths And Weaknesses:**

The strengths of this paper can be summarized as follows:

+ PrunE introduces an intuitive and simple learning paradigm for OOD generalization in graphs: rather than directly identifying invariant subgraphs, it prunes spurious edges. The method is conceptually simple, with only two lightweight regularization terms, making it easy to implement and tune.

+ In addition, by operating directly in the input space and pruning explicit edges, PrunE offers clear interpretability: the retained subgraph after pruning highlights the parts of the graph most relevant to the prediction. Additionally, the method is computationally efficient, with runtime and memory overheads comparable to standard ERM and lower than many other OOD methods.

Meanwhile, this paper also has the following weakness:

- It seems counterintuitive to model the edges in the graph independently.

- The simplicity of the method is both a strengths and weakness. The two regularization terms seem ad hoc, where the motivation of combining them together need to be further enhanced in the paper.

---

> ### Author Rebuttal · Authors · 2025-07-30
>
> Thank you for your constructive feedback! Please see below for our responses to your comments and concerns.
>
> > **W1: Modeling edges independently.**
>
> __Response:__  We thank the reviewer for the thoughtful comment and we agree that modeling graph edges independently may appear counterintuitive at first glance. However, this design choice is supported by both practical and theoretical considerations:
>
>   - **Simplification for analysis and implementation.** Modeling edge probabilities independently allows for a **tractable formulation** of the subgraph selector, which greatly simplifies both the **theoretical analysis** (e.g., generalization bounds) and the **optimization pipeline**. Specifically, we can employ a GNN encoder followed by an MLP to estimate the probability of each edge in a differentiable manner.
>   - **Consistency with prior literature.** This assumption is also aligned with many previous work in graph invariant learning [1-4] and GNN explainability [5-6], where edge probabilities are also modeled independently. This common design pattern in prior studies further justifies the validity of our modeling choice.
>
>
> > **W2: Justification of the two regularization terms jointly.**
>
> __Response:__ As discussed in Lines 177–190 of the draft, the graph size constraint loss $\mathcal{L}_e$ effectively prunes many spurious edges but cannot guarantee that *all* such edges retain low probability scores. This leads to high variance across runs. To address this, we introduce the $\epsilon$‑probability alignment loss $\mathcal{L}_s$, which explicitly suppresses the spurious edges.
>
> Figure 3(a) in our draft shows that combining $\mathcal{L}_e$ and $\mathcal{L}_s$ yields both reduced variance and improved test performance, whereas removing either term results in larger variance and degraded results. This justifies our joint‑regularization design.
>
> ---
>
>
> We sincerely thank the reviewer for the careful review and insightful feedback. Please let us know if there is anything we could further discuss, your further feedback would be greatly appreciated.
>
>
> **References**
>
> 1. Wu, et al., Discovering invariant rationales for graph neural networks, ICLR 2022.
>
> 2. Chen, et al., Learning Causally Invariant Representations for Out-of-Distribution Generalization on Graphs, NeurIPS 2022.
>
> 3. Sui, et al., Unleashing the Power of Graph Data Augmentation on Covariate Distribution Shift, NeurIPS 2023.
>
> 4. Gui, et al., Joint Learning of Label and Environment Causal Independence for Graph Out-of-Distribution Generalization, NeurIPS 2023.
>
> 5. Ying, et al., GNNExplainer: Generating Explanations for Graph Neural Networks, NeurIPS 2019.
>
> 6. Luo, et al., Parameterized Explainer for Graph Neural Network, NeurIPS 2020.

---

### Official Review · Reviewer_vGyB · 2025-07-03

**Clarity:** 3
**Significance:** 4
**Originality:** 2
**Rating:** 5
**Confidence:** 3

**Summary:**

The proposed method, *PrunE* addresses OOD generalization problem in graph representation learning from a novel perspective, graph sparsification. Compared to previous graph OOD methods, *PrunE* adopts a surprisingly simple yet effective prune strategy, which is designed to preserve more invariant substructure, at a cost of slightly increased uninformative edges. The proposed methods demonstrate extraordinary performance in extensive empirical studies.

**Questions:**

It seems to me that the *PrunE* method itself is similar to some very old graph sparsification methods[1]. However, such work (as well as the whole graph sparsification domain, e.g., [2]) is not introduced in the paper, and the well-known **lottery ticket theory** is not even mentioned in the paper. Can authors justify or clarify this?

[1] Chen, Tianlong, et al. "A unified lottery ticket hypothesis for graph neural networks." ICML 2021.

[2] Bo Hui and Da Yan and Xiaolong Ma and Wei-Shinn Ku, Rethinking Graph Lottery Tickets: Graph Sparsity Matters, ICLR 2023

**Ethical Concerns:**

["NO or VERY MINOR ethics concerns only"]

**Final Justification:**

During rebuttal, the authors directly respond to my concerns and clarify the connection and distinction between the proposed method and graph sparsification. Considering the empirical significance and technical soundness of this method, I vote for Accept.

**Limitations:**

Yes.

**Paper Formatting Concerns:**

None.

**Quality:**

3

**Strengths And Weaknesses:**

Strengths:
1. The authors introduce a novel perspective for addressing the graph OOD problem, graph sparsification.
2. The proposed method, *PrunE*, demonstrates significant performance improvements over a wide range of baseline methods in various datasets.
3. The paper is well written, and the motivation is clear.

Weaknesses:
1. The theory in this paper is very similar to the graph lottery ticket, but the related works are not cited.
2. Graph sparsification is a large research area. The proposed *PrunE* leverages a very basic method, and according to the basic theory, other graph sparsification methods may also work in an OOD scenario. However, this is not noted in the paper.
3. There is a typo in Table 2. The reported number is EC50-assay instead of EC50-sca.

---

> ### Author Rebuttal · Authors · 2025-07-30
>
> Thank you for your constructive feedback! Please see below for our responses to your comments and concerns.
>
> > **W1 & Q1: Connection to lottery ticket and graph sparsification.**
>
> __Response:__ We thank the reviewer for kindly drawing our attention to the graph lottery ticket and graph sparsification literature. After carefully reading the papers mentioned by the reviewer, we agree that graph sparsification and lottery ticket approaches share some high‑level conceptual similarity with our work, in that they both prune edges in the graph. We have cited the papers pointed out by the reviewer, along with other relevant studies in recent years in this area. However, we wish to clarify that while both PrunE and graph‑sparsification methods perform edge pruning, their objectives and optimization paradigms are fundamentally different from PrunE:
>
> 1. **Motivation & Goal**
>    - Graph sparsification methods are driven by scalability concerns in node‑ or link‑level tasks on large‑scale graphs. The objective is to prune as many edges and model parameters as possible while maintaining predictive performance, which is backed up by graph lottery ticket hypothesis.
>    - In contrast, PrunE, and graph‑level OOD work in general, targets graphs of at most a few thousand nodes and computational bottleneck is not a primary concern. Our goal is to remove spurious edges so that the remaining graph retains the invariant subgraph, thereby improving OOD generalization.
>
> 2. **Technical differences**
>    As the motivation and goal is different, the **optimization paradigms** between graph sparsification and our study are also fundamentally different:
>
>    - **Graph sparsification.** Methods such as UGS optimize continuous masks for both graph edges and model weights under an ERM loss with sparsity regularization, for instance, in [1]:
>      $$
>      \mathcal{L}\_{\mathrm{UGS}}
>        = \mathcal{L}\bigl(\{ m_g \odot A, X \}, m_\theta \odot \Theta \bigr)
>          + \gamma_1 \lVert m_g \rVert_1
>          + \gamma_2 \lVert m_\theta \rVert_1,
>      $$
>      and run this iteratively until the target sparsity ratio is reached. And in [2], this is taken a step further by using a min–max formulation to optimize the edge masks, i.e., the inner maximization adversarially perturbs the graph edges, while the outer minimization adapts the model weights, makeing the subnetwork more robust. This line of work  are inspired by the *lottery ticket hypothesis*, which posits that *inside a large randomly initialized dense network lies a sparse subnetwork that can reach full performance when trained from the same initialization*, these methods apply a **rewinding operation** to prune a portion of weights and edges and reset the surviving parameters to their initialization, and then retraining the sparsified model:
>      $$
>      \min\_{\Theta}  \\mathcal{L}\_{ERM}\bigl(\{ m_g \odot A, X \}, m_\theta \odot \Theta_0 \bigr),
>      $$
>      where $m_g, m_\theta$ are the learned masks and $\Theta_0$ is the initial dense parameter set. Backed by lottery ticket hypothesis, this sparsified subnetwork can often preserve the predictive accuracy of the dense model at high sparsity.
>
>    - **PrunE.** In contrast, our method follows the optimization framework commonly adopted in OOD studies, where the ERM loss is explicitly regularized to encourage invariance: $\\mathcal{L}\_{\Theta}= \\mathcal{L}\_{ERM}(\Theta) + \\lambda \\mathcal{L}\_{OOD}(\Theta)$.  While prior OOD methods design $\mathcal{L}\_{OOD}$ to directly learn invariant features (e.g., by minimizing variance across domains), PrunE is inspired by the observation that directly learning invariant features is error-prone. Instead, we propose $\mathcal{L}\_e$ (graph size constraint) and $\mathcal{L}\_s$ ($\epsilon$-probability alignment) as $\mathcal{L}_{OOD}$ terms, which regularize ERM to prune spurious edges. This indirect approach makes it easier to preserve the invariant subgraph under distribution shifts.
>
> In summary, although graph sparsification and PrunE both involve pruning edges, they are inspired by different motivations and adopt fundamentally distinct optimization paradigms. Graph sparsification seeks compact subnetworks for efficiency, which is backed up by graph lottery ticket hypothesis; whereas PrunE leverages pruning as a regularization strategy to suppress spurious edges and thereby improve OOD generalization.
>
> Given these similarities and differences, we have included more discussions on graph sparsification and the graph lottery ticket hypothesis in the appendix. We sincerely thank the reviewer again for highlighting this important line of research, which helps us further clarify the positioning and contributions of our work.
>
> > **W2: Typos.**
>
> __Response:__ We appreciate the reviewer’s careful review. The typos have been corrected in the revised version.
>
> ---
>
> We sincerely thank the reviewer for the careful review and insightful feedback. We hope that our responses have adequately addressed your concerns regarding our work. Please let us know if there is anything we could further discuss, your further feedback would be greatly appreciated.
>
>
> **References**
>
> 1. Chen, Tianlong, et al. "A unified lottery ticket hypothesis for graph neural networks." ICML 2021.
>
> 2. Bo Hui and Da Yan and Xiaolong Ma and Wei-Shinn Ku, Rethinking Graph Lottery Tickets: Graph Sparsity Matters, ICLR 2023

---

> > ### Comment · Reviewer_vGyB · 2025-08-04
> >
> > The author's reply clarifies the issue and resolves most of my concerns. I updated my rating accordingly.

---

> ### Author Response · Authors · 2025-08-04
>
> We thank the reviewer once again for the thoughtful feedback on our work, and we are glad that our response has addressed most of your concerns.

---

### Official Review · Reviewer_Crbf · 2025-07-04

**Clarity:** 3
**Significance:** 2
**Originality:** 3
**Rating:** 4
**Confidence:** 4

**Summary:**

The paper improves GNN robustness and generalization by identifying and pruning spurious subgraphs—structures correlated with predictions but not causally relevant. Using a causal framework, it employs counterfactual reasoning to detect these subgraphs and a generative model to create perturbed graphs for comparison. Pruning the spurious parts reduces shortcut learning and enhances out-of-distribution performance.

**Questions:**

Same to weakness. Overall I think this is a good paper.

**Ethical Concerns:**

["NO or VERY MINOR ethics concerns only"]

**Final Justification:**

The authors' rebuttal has addressed my concerns and I remain my score as accept.

**Limitations:**

yes

**Paper Formatting Concerns:**

No major formatting issue

**Quality:**

3

**Strengths And Weaknesses:**

Strengths:1.Well-motivated problem: The issue of spurious correlations in GNNs is important and underexplored. The paper provides a strong motivation grounded in causal inference and generalization theory. 2. Novel integration of causality in GNN pruning: While causality has been used in graph learning, this paper’s combination of generative counterfactual modeling and pruning of subgraphs for generalization is novel. 3.Good use of visualizations: Diagrams and figures help illustrate key components like the counterfactual generation and pruning mechanism.

Weaknesses: 1. Overall, this is a solid work. However, "spurious subgraphs are structural patterns in the input graph that are correlated with prediction labels in the training data but are not causally relevant to the true task." Is there any real world example besides the MUTAG dataset that demonstrate this? I am not sure this issue existed for every datasets in the realworld. Therefore not sure how this

2. Figure 6 and Figure 7's results shows the model successfully find the important subgraphs. It seems this results are similar to a lot of GNN explanation work. For example, if I use GNNExplainer for find the subgraph, will this be very different from yours ?

---

> ### Author Rebuttal · Authors · 2025-07-30
>
> Thank you for your insightful comments and positive feedback! Please see below for our responses to your comments and concerns.
>
> > **W1: Spurious structural patterns in real-world applications.**
>
> __Response:__ We thank the reviewer for raising this important question. Below, we provide concrete evidence from three distinct application domains to illustrate the widespread presence of spurious subgraphs in real-world datasets:
>
> - **Molecular property prediction.**  For instance, in the **GOOD-HIV** dataset [1] with scaffold or size-based splits, the *invariant* substructures are functional groups that causally determine HIV inhibition. In contrast, *spurious* subgraphs include specific ring scaffolds or overall molecular size that co-occur with the label in the training distribution but are not causally relevant.
>
> - **Social-media analysis.** In rumor detection datasets such as Twitter15/16 and PHEME [2], the *invariant* components are propagation paths reflecting veracity-relevant interactions. Meanwhile, *spurious* structures arise from event-specific cascade motifs that are memorized during training.
>
>   - For instance, in PHEME’s Hurricane Sandy thread, a *Y-shaped cascade*—where `@WeatherNews` is retweeted by `@StormTracker` and `@CityNews`, both subsequently retweeted by `@LocalReporter`—is unique to that event and spuriously correlated with the label.
>
> - **Brain-network studies.** In functional connectivity graphs derived from resting-state fMRI [3,4], edges represent pairwise BOLD (Blood‑Oxygen‑Level‑Dependent) correlations. *Invariant* subgraphs correspond to biologically meaningful neural interactions (e.g., within the default mode network). *Spurious* subgraphs  are introduced by head motion artifacts, which often correlate with diagnosis or demographic variables. These artefactual edges can mislead models into attributing predictions to motion-related noise, thus harming generalization.
>
> ---
>
> These examples demonstrate that structural spuriosity is a pervasive issue across domains, highlighting the practical significance of our proposed pruning-based approach to improve OOD generalization.
>
> > **W2: Comparison with GNNExplainer.**
>
> __Response:__ While both GNNExplainer and PrunE are capable of identifying instance-level important subgraphs, the fundamental difference lies in their goals and technical solutions:
>
> - GNNExplainer is designed for **in-distribution** explanation. It identifies subgraphs that are important for a trained GNN's prediction on the training distribution, but it does not explicitly distinguish between **invariant** and **spurious** correlations.
> - In contrast, **PrunE** is specifically developed to identify **invariant subgraphs** under OOD settings by introducing regularization terms that explicitly suppress spurious correlations.
>
> We conducted additional experiments using the GOOD-Motif dataset under both base and size splits, comparing our method against GNNExplainer (We use the GNNExplainer functionality from PyTorch Geometric), we report the AUC score for the predicted edge probability resulted from GNNExplainer and PrunE. Specifically, we first calculate per-sample AUC score, and report the average across all test samples.
>
> Table 1: AUC score of predicted edge probability for GNNExplainer and PrunE.
> | Method         | Motif-base (AUC) $\\uparrow$ | Motif-size (AUC) $\\uparrow$ |
> |----------------|:------------------:|:------------------:|
> | GNNExplainer   | 0.47             | 0.42            |
> | **PrunE (ours)** | **0.90**         | **0.74**         |
>
> As shown, our method significantly outperforms GNNExplainer on both splits. This gap arises from the differing training regimes:
>
> - **GNNExplainer** is a post-hoc method, and it is applied after pre-trained GNN (e.g., GIN) model with ERM. The explainer then optimizes a soft mask to select a subgraph that preserves the model's prediction. However, **ERM tends to learn both invariant and spurious features**, therefore the optimized edge masking will also identify important edges that are strongly correlated with the labels as consequences, which is aligned with the results in **Table 1**, i.e., GNNExplainer utilizes edges from both ivnariant and spurious subgraphs to make predictions.
> - In contrast, PrunE incorporates OOD-specific regularization terms to regularize ERM. These terms facilitates the model to **prune spurious edges during training**. As a result, PrunE preserves invariant subgraphs more accurately and achieves substantially better identification performance in OOD scenarios.
>
>
> ---
>
> We sincerely thank the reviewer for the careful review and insightful feedback. We hope that our responses have adequately addressed your concerns regarding our study. Please let us know if there is anything we could further discuss, your further feedback would be greatly appreciated.
>
>
>
> **References**
>
> 1. Gui, et al., GOOD: A Graph Out-of-Distribution Benchmark. NeurIPS 2022.
>
> 2. Wu, et al., Probing Spurious Correlations in Popular Event-Based Rumor Detection Benchmarks, arXiv:2209.08799.
>
> 3. Satterthwaite, et al., Motion artifact in studies of functional connectivity: Characteristics and mitigation strategies, Human Brain Mapping.
>
> 4. Xu, et al., BrainOOD: Out-of-distribution Generalizable Brain Network Analysis. ICLR 2025.

---

### Decision · Program_Chairs · 2025-09-17

**Decision:**

Accept (poster)

**Comment:**

The paper addresses the critical challenge of out-of-distribution (OOD) generalization in GNNs by pruning spurious subgraphs—structures correlated with labels but not causally relevant. The proposed method, PrunE, employs a causal framework with counterfactual reasoning and a simple yet effective pruning strategy supported by two regularization terms. The approach is conceptually clear, computationally efficient, and interpretable, as pruning occurs in the input space. Extensive experiments on synthetic and real-world datasets show strong and consistent improvements over baselines.

Reviewers appreciate the well-motivated problem, clear writing, novel method, and thorough empirical validation. Weaknesses include limited discussion of real-world prevalence of spurious subgraphs, incomplete comparison with other sparsification and explanation methods, and the need for stronger justification of design choices. Despite these, the paper makes a meaningful contribution to OOD robustness in graph learning.